



# Constraining the response factors of an extractive electrospray ionization mass spectrometer for near-molecular aerosol speciation

Dongyu S. Wang[1,*], Chuan Ping Lee[1], Jordan E. Krechmer[2], Francesca Majluf[2], Yandong Tong[1], Manjula R. Canagaratna[2], Julia Schmale[1,3], André S.H. Prévôt[1], Urs Baltensperger[1], Josef Dommen[1], Imad El
Haddad[1,*], Jay G. Slowik[1,*], and David M. Bell[1,*]

[1]Laboratory of Atmospheric Chemistry, Paul Scherrer Institute, Villigen PSI, Aargau, 5232, Switzerland
[2]Center for Aerosol and Cloud Chemistry, Aerodyne Research Inc., Billerica, Massachusetts, 01821, United States
[3]Extreme Environments Research Laboratory, École Polytechnique Fédérale de Lausanne, Sion, Valais, 1951, Switzerland

*Corresponding authors

**Abstract.** Online characterization of aerosol composition at the near-molecular level is key to understanding chemical reaction mechanisms, kinetics, and sources under various atmospheric conditions. The recently developed extractive electrospray ionization time-of-flight mass spectrometer (EESI-TOF) is capable of detecting a wide range of organic oxidation products in the particle phase in real time with minimal fragmentation. Quantification can sometimes be hindered by a lack of available commercial standards for aerosol constituents, however. Good correlations between the EESI-TOF and other aerosol speciation
techniques have been reported, though no attempts have yet been made to parameterize the EESI-TOF response factor for different chemical species. Here, we report the first parameterization of the response factors of the EESI-TOF for secondary organic aerosol (SOA) at the near-molecular level based on their elemental composition. SOA was formed by ozonolysis of monoterpenes or OH-oxidation of aromatics inside an oxidation flow reactor (OFR) using ammonium nitrate as seed particles. A Vocus proton-transfer reaction mass spectrometer (Vocus-PTR) and a high-resolution aerosol mass spectrometer (AMS)
were used to determine the gas phase molecular composition and the particle phase bulk chemical composition, respectively. The EESI response factors towards bulk SOA and the inorganic coating were constrained by intercomparison with the AMS. The highest bulk EESI response factor was observed for SOA produced from 1,3,5-trimethylbenzene, followed by those produced from $d$-limonene and $o$-cresol, consistent with previous findings. The near-molecular EESI response factors were derived from intercomparisons with Vocus-PTR measurements, and were found to vary from $10^3$ to $10^6$ ions s$^{-1}$ ppb$^{-1}$, mostly
within $\pm 1$ order of magnitude of their geometric mean of $10^{4.5}$ ions s$^{-1}$ ppb$^{-1}$. For aromatic SOA components, the EESI response factors correlated with molecular weight and oxygen content, and inversely correlated with volatility. The near-molecular response factors agreed within a factor of 20 for isomers observed across the aromatics and biogenic systems. Parameterization of the near-molecular response factors based on the measured elemental formulae could reproduce the empirically determined response factor for a single VOC system to within a factor of 5 for the configuration of our mass spectrometers. Results
demonstrate that standard-free quantification using EESI-TOF is possible.



## 1. Introduction

Suspended particulate matter, or aerosol, is ubiquitous in the troposphere with far-reaching implications for public health, air quality, and climate (Jimenez et al., 2009; Dockery et al., 1995). The aerosol composition can have large spatiotemporal variations, evolving over the course of a week or changing drastically within a matter of seconds, depending on the emission source, meteorology, and atmospheric chemistry. Large discrepancies have been reported between ambient observations and modelling results (Volkamer et al., 2006; Tsigaridis et al., 2014). Real-time aerosol speciation is therefore required to temporally resolve and understand aerosol dynamics. To this end, aerosol mass spectrometer (AMS) using flash-vaporization and electron impact (EI) ionization serves as a reliable quantification method to determine the bulk composition of $PM_1$ or $PM_{2.5}$ (i.e. particles with an aerodynamic diameter <1 μm or <2.5 μm, respectively) over long periods of time both online (DeCarlo et al., 2006; Jimenez et al., 2009; Ng et al., 2011) and offline (Daellenbach et al., 2016). However, the extensive thermal and EI-induced fragmentations render the technique ill-suited to infer the molecular identity of individual components, with very few exceptions (Alfarra et al., 2007; Budisulistiorini et al., 2013). More recent techniques such as the Filter Inlet for Gases and AEROsols (FIGAERO, Lopez-Hilfiker et al., 2014) and the Chemical Analysis of Aerosol Online (CHARON) inlet (Müller et al., 2017; Eichler et al., 2015) utilize chemical ionization mass spectrometry (CIMS) instead. Although CIMS is a much "softer" ionization technique than EI, ionization-induced fragmentation still occurs to some extent depending on the analyte, e.g. during proton-transfer reactions (PTR) used by a CHARON (Leglise et al., 2019; Murschell et al., 2017; Duncianu et al., 2017). The need for thermal volatilization to convert the particles to vapors before ionization may introduce artefacts from the decomposition of thermally labile compounds (Leglise et al., 2019; Stark et al., 2017).

In contrast, the extractive electrospray ionization (EESI) mass spectrometry (MS) can be used for continuous online aerosol analysis without sample preparation (Gallimore et al., 2013; Chen et al., 2006; Lopez-Hilfiker et al., 2019; Doezema et al., 2012), reducing associated artefacts. Electrospray ionization (ESI) is known to be a soft ionization technique, capable of preserving even non-covalent interactions, e.g. protein-protein interactions (Siuzdak et al., 1996), though some fragmentation reactions have been reported to occur within the ESI droplet (Rovelli et al., 2020). In EESI, charged droplets generated by an electrospray (ES) collide with the analyte aerosols. A denuder is used to strip gas-phase species and reduce measurement interference. Soluble particulate analytes are extracted by the charged droplets and ionized via Coulomb explosion as the charged droplets rapidly evaporate (Kebarle and Peschke 2000). The addition of sodium iodide (NaI) to the ES solution as a dopant suppresses unwanted ionization pathways (e.g. $H^+$ transfer), resulting in predominately the formation of $Na^+$ adduct, $[M+Na]^+$. With this ionization scheme, the recently developed EESI time-of-flight mass spectrometer (EESI-TOF) is able to achieve detection limits on the order of 1-10 ng $m^{-3}$ for compounds like raffinose and dipentaerythritol on the timescale of 5 s (Lopez-Hilfiker et al., 2019). The low detection limit and fast time resolution of the EESI-TOF have enabled real-time near-molecular speciation of organic aerosol (i.e., identification of the chemical formulae of molecular ions) for various laboratory and field applications, both indoors and outdoors (Brown et al., 2021; Liu et al., 2019; Pagonis et al., 2021; Pospisilova et al., 2020; Qi et al., 2019; Stefenelli et al., 2019). Further adaptation of the EESI with the Orbitrap mass analyzer





potentially allows for structural elucidation using online tandem mass spectrometry, in addition to unambiguous assignment of the chemical formulae (Lee et al., 2020).

Tests with authentic standards show that EESI-TOF can quantify target inorganic and organic analytes present in complex sample matrices (Fang et al., 2016; Giannoukos et al., 2020; Lopez-Hilfiker et al., 2019; Wu et al., 2013). However, the absolute and relative ionization efficiencies (RIE) of EESI or ESI (where the analyte is directly infused into the ES solution) towards different compounds could vary by orders of magnitude depending on the instrument setting, sample matrices,
electrospray solutions, and operators (Kruve et al., 2013; P. Liigand et al., 2018; Kruve et al., 2014; Oss et al., 2010; Mayhew et al., 2020; Lopez-Hilfiker et al., 2019). For instance, the EESI response factors for two carboxylic acids, citric acid ($C_6H_8O_7$) and azelaic acid ($C_9H_{16}O_4$) differ by 1-2 orders of magnitude when detected as $Na^+$ adducts ([M+Na]$^+$) using a 50/50 methanol/water ES solvent spiked with 100 ppm NaI (Lopez-Hilfiker et al., 2019). Similarly, the RIE of [M+Na]$^+$ varies by 4 orders of magnitude for a selection of 19 standard compounds detected with ESI using a 80/20 acetonitrile/ 0.1 M sodium
acetate solution (Kruve et al., 2013).

In this study, we estimate the EESI-TOF response factor to various particle-phase species found in biogenic and anthropogenic SOAs. In lieu of authentic standards for SOA, which are complex mixtures comprised of hundreds of unique compounds, we compare the EESI-TOF measurements with established quantitative and semi-quantitative techniques to constrain the bulk and near-molecular response factors. Our results show that the EESI-TOF response factor to SOA
components varies from $10^3$ to $10^6$ ions s$^{-1}$ ppb$^{-1}$ with structural dependences, as evidenced in the differences for isomers observed across multiple systems. In general, response factors were observed to increase with the analyte molecule size and oxygen content. Based on properties derived from the measured elemental formula, regression models could predict the response factor of individual ions to within a factor of 5 of the measured values for a single component system.

## 2. Experimental Setup

### 2.1 Oxidation flow reactor

Oxidation reactions took place inside a Pyrex oxidation flow reactor (OFR) with an inner diameter of 7.4 cm and a length of 104 cm, which has been described previously (Molteni et al., 2018). Instruments sampled from the outlet of the OFR. The total flow rate inside the OFR was maintained at 12 L min$^{-1}$, resulting in a plug flow velocity of 0.0465 m s$^{-1}$. Excess flow was vented to avoid overpressure in the OFR. Milliliters of α-pinene (Sigma-Aldrich, ≥99%), limonene (Sigma-Aldrich, 97%),
o-cresol ("cresol", Sigma-Aldrich, ≥99%), or 1,3,5-trimethylbenzene ("TMB", Sigma-Aldrich, 98%) were placed inside a glass vial connected orthogonally to a dry clean air carrier flow to supply VOC precursors. Dry clean air and VOC precursors were injected near the entrance region of the OFR (x = 0 cm), whereas ozone ($O_3$) was injected further downstream (x = 72 cm) coaxially into the center of the OFR, as illustrated in Figure S1, resulting in an effective oxidation time of 8.4 s. $O_3$ was generated using a mercury lamp with dry clean air. For experiments with aromatic precursors, tetramethylethylene (TME,
Sigma-Aldrich, ≥ 99%) was injected together with the VOC precursors to generate OH radicals via TME ozonolysis. In the





absence of TME (and thereby OH), ozonolysis of aromatics is negligible (Atkinson and Arey 2003). VOC and $O_3$ injection rates were adjusted to minimize nucleation inside the OFR. When it was necessary to promote condensation of organic vapors, ammonium nitrate ($NH_4NO_3$) seed particles were generated from 0.0025 to 0.3 M aqueous solutions using a nebulizer, dried by a rubin silica gel diffusion dryer, and injected near the entrance of the OFR. A multichannel activated charcoal denuder was used to remove possible gas-phase contaminants generated during ammonium nitrate nebulization. The particle count mean diameter ranged from 60 to 100 nm during seed injections.

## 2.2 Gas-phase quantification

$O_3$ concentration was measured using an $O_3$ monitor (Thermo 49C). VOC precursors and oxidation products were measured using a Vocus-PTR long time-of-flight mass spectrometer (Vocus-PTR, Tofwerk, AG). The design and operations of the (Vocus-)PTR are described in detail elsewhere (Krechmer et al., 2018; Yuan et al., 2017). Ionization of an analyte, M, occurs via proton transfer with the $H_3O^+$ reagent ion inside the PTR drift tube if M has a higher proton affinity than $H_3O^+$:

$$H_3O^+ + M \leftrightarrow [M + H]^+ + H_2O \qquad \text{Eq. (1)}$$

Depending on the drift tube setting and the proton affinity of M, the reaction in Eq.(1) is reversible. If $H_3O^+$ depletion is negligible, the abundance of $[M+H]^+$ ion scales linearly with the concentration of M with a slope equal to the product of the species-dependent reaction rate constant $k_{MH}$, drift time, $\Delta t$, and the $H_3O^+$ reagent ion abundance. The proton-transfer-reaction is exothermic and the ionization-induced fragmentation occurs to varying degrees (~0 to ~100 %) as the result of dehydration, $H_2$ elimination, alkyl group loss, or $HNO_3$ loss (Yuan et al., 2017; Duncianu et al., 2017; Leglise et al., 2019), which can be represented by the term, $F_{MH+}$. The observed intensity of any ion is also a function of the ion abundance, $I_{MH+}$ and ion transmission efficiency, $T_{MH+}$. All together, the sensitivity for an analyte normalized to a $H_3O^+$ signal of $10^6$ ions s$^{-1}$ (cps) is expressed as follows (Sekimoto et al., 2017; Cappellin et al., 2012; Yuan et al., 2017),

$$Sensitivity = N \times 10^{-3} \times \Delta t \frac{T_{MH+}}{T_{H_3O^+}} F_{MH+} \times k_{MH} \qquad \text{Eq. (2)}$$

where $N$ is the number density of gas inside the drift tube. The overall ion transmission efficiency is a function of the mass-to-charge ratio ($m/z$) as determined by the ion optics and TOF extraction duty cycle. For the Vocus-PTR, the mass transmission efficiency was fitted to a lognormal function, as shown in Figure S2, leveling out at approximately 0.175 relative to the maximum transmission at ~95 $m/z$. The reaction rate coefficient is dependent on the polarizability, $\alpha$, and the permanent dipole moment of the analyte, $\mu_D$. For similarly functionalized compounds, $\alpha$ linearly correlates with the molecular weight (MW) whereas $\mu_D$ is independent of MW, respectively (Sekimoto et al., 2017). Given the molecular weight and the elemental composition, consisting of only carbon (C), hydrogen (H), and oxygen (O) for the oxidation products of interest in the present study, the $k_{MH}$ values estimated using parameterization should agree within 50% of the measured values, provided that the ion transmission efficiency and fragmentation behavior for the analyte is known (Sekimoto et al., 2017). Since this parameterization was derived using non-oxygenated or lightly oxygenated species (i.e. number of oxygen $n_O \leq 2$), the estimated


$k_{MH}$ values would become more uncertain as we apply the parameterization for more oxygenated species. In addition, $F_{MH}$ is known to vary with functionalization and, in general, increase with the size of the molecule (Yuan et al., 2017), which may result in differing sensitivities for isomers and more underestimated sensitivities for larger molecules, respectively.

In this study, the Vocus-PTR core-sampled at 0.1 L min$^{-1}$ from a 4 L min$^{-1}$ of air drawn from the OFR via a ¼ inch (outer diameter, OD) Teflon line ~ 0.2 m in length. The Vocus-PTR differs from traditional PTR in its reagent ion source design, as well as the use of quadrupole-based focusing ion-mole reactor (fIMR) instead of a standard drift tube (Krechmer et al., 2018). Comparison of the Vocus-PTR with traditional Q- and TOF-PTR shows that it has much better sensitivity (1-3 orders of magnitude), with the estimated and measured sensitivities mostly agreeing within a factor of 2 (Holzinger et al.,

2019). The Vocus fIMR reactor was heated and maintained at 60 °C, with an axial gradient of 330 V, and a pressure of 2.5 mbar. The reduced electric field (E/N) value is estimated to be 59 Td. Vocus-PTR data was analyzed using Tofware 3.2 (Tofwerk, AG) in Igor Pro 8.

## 2.3 Particle-phase quantification

A high-resolution time-of-flight aerosol mass spectrometer (AMS, Aerodyne Research Inc.) was used to measure the

bulk chemical composition of non-refractory (NR) aerosol (Canagaratna et al., 2007; DeCarlo et al., 2006). Sample was drawn at 1.27 cm$^3$ s$^{-1}$ through a 100 µm critical orifice and then focused by a PM$_{2.5}$ aerodynamic lens (Peck et al., 2016; Williams et al., 2013; Xu et al., 2017). The AMS operated in mass spectrum (MS) mode with 1 min time resolution, where the chopper alternated between the closed (i.e. particle beam blocked) and the open positions (i.e. particle beam is unobstructed) to produce the difference spectra for aerosol quantification. The ionisation efficiencies of ammonium, (NH$_4^+$), nitrate (NO$_3^+$) and sulfate

(SO$_4^+$) ions were calibrated using 300 nm size-selected NH$_4$NO$_3$ and (NH$_4$)$_2$SO$_4$ aerosol at the beginning and end of the campaign by following the standard protocols (Canagaratna et al., 2007; DeCarlo et al., 2006). The relative ionization efficiencies for NH$_4^+$ and SO$_4^+$, RIE$_{NH4+}$ and RIE$_{SO4+}$ were measured to be 4.01 and 1.24, respectively. Default relative ionization efficiency is assumed for organics at 1.4 without applying any correction based on the average oxidation state of carbon (OS$_C$). AMS data were analysed in Igor Pro 6.37 (Wavemetrics, Inc.) using the Squirrel (version 1.60G) and Pika

(1.20G) analysis packages. When sampling high concentrations of inorganic salts (e.g. NH$_4$NO$_3$), CO$_2^+$ ions could be produced from organic residues at the vaporizer surface, leading to overestimation of the organic aerosol (OA) mass (Pieber et al., 2016). Characterization and correction of the vaporizer artefacts are shown in Figure S3. The particle size distribution was monitored using a scanning mobility particle sizer (SMPS 3938, TSI). The total particle mass concentration is calculated from the total particle volume concentration measured by the SMPS assuming that OA-coated seed particles, which comprise predominately

(>90%) of NH$_4$NO$_3$, has the same density as pure NH$_4$NO$_3$ aerosol at 1.69 g cm$^{-3}$ (Sarangi et al., 2016). To account for the AMS collection efficiency, the OA concentration is calculated by multiplying the OA mass fraction as measured by the AMS (after accounting for the vaporizer effect) with the total particle mass concentration as calculated from SMPS measurements.



## 2.4 Extractive electrospray ionization

An extractive electrospray ionization mass spectrometer, EESI-TOF (Tofwerk AG) was used for online, near-
molecular organic aerosol speciation (Lopez-Hilfiker et al., 2019). The mass resolving power ($m/\Delta m$) was approximately 4200
at $m/z$ 200. The electrospray (ES) was generated from a 1:1 water:acetonitrile solution containing 100 ppm NaI dopant
delivered through a fused-silica capillary with 150-250 mbar backing pressure at an electrical potential of 2.7-2.9 kV relative
to the MS atmospheric pressure interface. The ion capillary at the MS interface was heated to 270 °C to facilitate ES droplet
evaporation for analyte ionization. Whereas adduct formation with metal cations obfuscate analyses by traditional (+)ESI
techniques that rely on the [M+H]$^+$ pathway, the intentional use of NaI for EESI ensures that Na$^+$-adduct formation ([M+Na]$^+$)
is the dominant ionization pathway for organic aerosol components. This makes the EESI technique more robust against salt
impurities (e.g. from aerosol samples, glassware, or solution) and ionization-induced fragmentations as compared to the
[M+H]$^+$ pathway (Kruve et al., 2013). In addition to [M+Na]$^+$ ions, acetonitrile-Na$^+$ adducts (i.e. [M+C$_2$H$_3$NNa]$^+$) were
observed in the mass spectra, along with negligible amounts of water-Na$^+$ adducts (i.e. [M+Na+H$_2$O]$^+$) and [M+H]$^+$ ions.
Analytes with labile hydrogen atoms, such as carboxylic acids or amines, could produce [M-nH+nNa+Na]$^+$ ions, where $n$
corresponds to the number of labile hydrogen atoms exchanged with Na. We expect the abundance of [M-nH+nNa+Na]$^+$
relative to [M+Na]$^+$ to be minor based on testing with standards (i.e. < 10% for *cis*-pinonic acid). The NH$_4$NO$_3$ seed particles
were detected as [NaNO$_3$+Na]$^+$ by the EESI-TOF.

Samples were drawn at 1 L min$^{-1}$ through a multi-channel extruded carbon denuder with a > 99.6% gas-phase removal
efficiency (e.g. for pinonic acid) (Tennison 1998) placed immediately upstream of the ionization region. A manifold was
placed upstream of the carbon denuder to direct the sample flow through either a high efficiency particulate air filter (HEPA,
Pall Corporation) to determine the background or a straight 10 mm OD stainless steel tube to measure the aerosol composition.
The manifold performed automated switching between the filtered and direct sampling periods, which lasted 1 and 5 min,
respectively. For an analyte, $x$, the interconversion of the mass concentration, $Mass_x$ and the background-corrected analyte ion
intensity, i.e the measured ion flux to the detector, $I_x$ depends on several factors as described in Lopez-Hilfiker et al. (2019):

$$Mass_x = I_x \left( \frac{MW_x}{EE_x \cdot CE_x \cdot IE_x \cdot TE_{m/z}} \right) \cdot \frac{1}{F} \qquad \text{Eq. (3)}$$

where $F$ is the inlet flow rate at 1 L min$^{-1}$, $MW_x$ is the molecular weight of the neutral species $x$, $EE_x$ is the liquid-phase
extraction efficiency inside the ES droplet, $CE_x$ is the collection efficiency of ES droplets by the ion capillary, $IE_x$ is the
ionization efficiency during the ES evaporation process, and $TE_{m/z}$ is the mass transmission efficiency, which depends on the
ion optics settings. For simplicity, the 4 efficiency factors are jointly expressed by the response factor ($RF_x$) instead,

$$Mass_x = I_x \left( \frac{MW_x}{RF_x} \right) \cdot \frac{1}{F} \qquad \text{Eq. (4)}$$

To determine the EESI-TOF $RF_x$ on the near-molecular level, we estimated the concentration of condensed organic
compounds during each seed injection, $P_{cond}$, based on the observed decrease in the gas-phase mixing ratios (in parts-per-





billion, ppb) as measured by the Vocus-PTR. For consistency and ease of comparison with the Vocus-PTR, we define a

response factor, $RF^*_x$ in terms of ions s$^{-1}$ ppb$^{-1}$

$$RF^*_x = \frac{I_x}{P_{cond,x}}$$    Eq. (5)

The $RF_x$ in ions molecule$^{-1}$ (Eq. 4) can be converted to $RF^*_x$ in ions s$^{-1}$ ppb$^{-1}$ (Eq. 5) using the instrument flow rate (~1 L min$^{-1}$ in this study) and the definition of the mixing ratio (1 ppb ≈ 2.46 x 10$^{10}$ molecules cm$^{-3}$ at 1 atm and 298 K).

**2.5 Gas-particle partitioning**

In the absence of vapor wall loss, the expected condensed concentration of an analyte $x$ during seed particle injection, $P_{cond,x}$ is equal to the decrease in its gas-phase concentration, $\Delta Gas_x$, increasing with the condensation sink ($CS$), which can be calculated from the observed particle size distribution

$$CS = 2\pi D \sum_i \beta_i d_{p,i} N_i$$    Eq. (6)

where $D$ is the gas diffusivity in m$^2$ s$^{-1}$, $\beta_i$ is the Fuchs-Sutugin correction factor for gas-phase diffusion over particles in the

transition regime for particle in the $i$th size bin, $d_{p,i}$ is the particle diameter of bin $i$, and $N_i$ is the particle number in bin $i$. In estimating $\beta i$, the mass accommodation coefficient was assumed to be unity (Kulmala et al., 2001). The Knudsen number ($Kn$) is estimated based on the pressure normal mean free path, $\lambda_P$ (Tang et al., 2015). The gas diffusivity estimated using Fuller's method (Fuller et al., 1966; Tang et al., 2015) ranges from 1.18·10$^{-5}$ to 3.69·10$^{-6}$ m$^2$ s$^{-1}$ for $C_3H_6$ and $C_{20}H_{32}O_{16}$, respectively. A list of $Kn$, $\beta$, $D$ values estimated assuming a particle diameter of 100 nm is shown in Table S1 for selected CH and CHO

compounds. A mean $D$ value of 6·10$^{-6}$ m$^2$ s$^{-1}$ is assumed for the $CS$ calculation.

In practice, the vapor wall loss is non-negligible, and the $P_{cond,x}$ is expected to be higher than the observed $\Delta Gas_x$. The effect of the vapor wall loss rate on the gas-particle partitioning behavior during each seed injection is modeled using KinSim v4.05 (Peng et al., 2019) in Igor Pro 8. For simplicity, generic oxidation products of varying assumed saturation vapor concentrations (i.e., 10$^{-2}$ μg m$^{-3}$ ≤ $C^*$ ≤ 10$^6$ μg m$^{-3}$) were used for the model in lieu of explicit representations of the oxidation

chemistries for all three VOC systems, which are beyond the scope of the current study (see Section S5 in the Supplementary Information for further details). For gas and/or particle species observed by the Vocus-PTR and/or EESI-TOF, the saturation vapor concentration at room temperature, $C^*$(298 K) is estimated using the molecular corridor approach (Li et al., 2016), based on the framework developed originally for the two-dimensional volatility basis set (Donahue et al., 2011),

$$\log_{10} C^*(298K) = (n_C^0 - n_c)b_c - n_O b_O - 2\frac{n_C n_O}{n_c + n_O} b_{CO}$$    Eq. (7)

where $n_C^0$ is the reference carbon number; $n_C$ and $n_O$ are the number of carbon and hydrogen atoms, respectively; $b_C$ and $b_O$ are the corresponding parameterization values for a specific compound class (e.g. CHO); $b_{CO}$ is the coefficient of carbon-oxygen non-ideality, $n_c n_O/(n_c + n_O)$, hereafter referred to as $NI_{CO}$. Note that terms involving nitrogen- and sulfur-containing compounds are not shown as the current work focuses on oxidized organic compounds (i.e. CHO) only. For CHO compounds, the $n_C^0$, $b_C$,





$b_O$, and $b_{CO}$ values are 22.66, 0.4481, 1.656, and -0.779, respectively (Li et al., 2016). For highly oxygenated molecules (HOM),

alternative $n_C^0$, $b_C$, $b_O$, and $b_{CO}$ values, e.g. 25, 0.475, 0.2, and 0.9 respectively, have been proposed to better account for the abundance of –OOH functional groups, which decreases the saturation vapor pressure less than for combined –OH and =O functional groups (Mohr et al., 2019; Tröstl et al., 2016; Pankow and Asher 2008). To accommodate the large range of oxidation products observed (e.g. $C_xH_yO_n$ where $n$=1-9) in the particle phase which span from HOM to singly oxygenated molecules, the formulation derived by Li et al. (2016) is applied to all molecular formulae.

**2.6 Estimation and regression analysis of $RF_x^*$**

Regression analyses on the logarithm of RIE of Na$^+$ adducts observed by (+) ESI suggest that ion-dipole interactions and chelation enhances Na$^+$ adduct formation efficiency (Guo et al., 1989; Kruve et al., 2013). The logarithm of RIE is also observed to increase as the logarithm of vacuum-solvent partitioning coefficient becomes more negative (Kruve et al., 2013). Insights into the determining factors in the $RF_x^*$ of Na$^+$-EESI-TOF may be gained through similar analysis. Because the Vocus-

PTR and EESI-TOF could only determine the molecular formula of the analytes, regression of $RF_x^*$ was performed using the elemental composition (i.e. $n_C$, $n_H$, $n_O$) and their derivative properties as features, such as the carbon-oxygen non-ideality, $NI_{CO}$ (Eq. 7), double bond equivalents ($DBE$, Eq. S16), and aromaticity ($X_c$, Eq. S17). Feature values were standardized (i.e. subtraction by the mean values and divided by the standard deviation) prior to fitting so that their relative importance can be compared based on their fitting coefficients. In total, 15 potential features were included in the initial analysis using 6 different

regressors from the scikit-learn packages in Spyder 4.1.4 and Python 3.8.3. Note that the model performance does not improve monotonically with the number of features used. The optimal set of features for each model was identified using default model hyperparameters to achieve a balance between goodness of fit (i.e. higher coefficient of determination, $R^2$, Eq.S21) and model complexity (i.e. lower number of features). Results obtained using linear ridge regression (LRR) and gradient boosting regression (GBR) are discussed in the main text. LRR instead of ordinary least square (OLS) regression was chosen to handle

multicolinearity and overfitting issues during regression with the use of $L_2$ regularization, i.e. by adding a term proportional to the sum of the square of feature weights ($||w||_2^2$) to the cost function. Nonparametric, decision-tree type regressor such as GBR was chosen because it is expected to be better than linear regressors at handling unforeseen interactions in the feature space, where an exhaustive search of all possible non-linear combinations of features is not feasible. Studies with ESI on the RIE of [M+H]$^+$ ions have reported significant improvement in prediction ability by random forest regression models (e.g. within ±1

order of magnitude of the expected values) compared to multiple linear regression models (e.g. within ±2 orders of magnitude of the expected values) for calibration standards (Liigand et al., 2020; Mayhew et al., 2020). See Section 7 in the S.I. for further details on the feature selection, model cross-validation, and model selection.



# 3. Results and Discussion

## 3.1 Near-molecular response factors

As shown for OH-oxidation of TMB, the EESI-TOF signals responded promptly to each seed injection, as indicated by the rising OA concentration and *CS* in Figures 1a and1c, respectively. During each seed injection, the EESI-TOF observed growth in signal from a wide range of oxidation products (e.g. $C_xH_yO_{1-9}$), with $C_9H_{14}O_{3-9}$ shown in Figure 1a. As shown in Figure 1c, decreases in gas-phase concentration during seed injections were observed by the Vocus-PTR at the same time as the increase in the EESI-TOF signals, though the change is hardly discernable under low *CS* conditions ($< 0.1$ s$^{-1}$) due to competition for

condensable species by the OFR wall. The SOA composition as measured by the EESI-TOF, averaged over all uptakes, is shown for cresol, TMB, and limonene oxidation products in Figure S5a, S5b, S5c, respectively. Significant contributions by $C_{10}$ and $C_{12}$ ions to the average EESI-TOF mass spectra were observed for SOA produced from OH oxidation of cresol and TMB, respectively, indicative of gas-phase dimer formation involving the $RO_2$ radical of TME (used as an OH source) via $RO_2+R'O_2$ cross reactions (Berndt et al., 2018). In contrast, the signal of $O_3$-limonene products was dominated by $C_{8-10}$ ions,

with negligible contribution from ions corresponding to dimers. Ions corresponding to small, volatile species (e.g. lightly oxidized $C_2$ to $C_5$ compounds) which are not expected in the particle phase were also observed, as shown in Figure S5a-c, which may be an indication of some extent of ion fragmentation with the EESI-TOF. A comparison of the major oxidation products, e.g. those having the same carbon number as the precursor VOC, between the EESI-TOF particle-phase and Vocus-PTR gas-phase measurements are shown in Figure 1b and Figure 1d for the TMB runs, as well as in Figure S6a-f for all VOC

systems studied. While EESI-TOF particle-phase measurements suggest that $C_9H_{14}O_x$ compounds are collectively much more abundant than either $C_9H_{12}O_x$ or $C_9H_{16}O_x$ compounds, Vocus-PTR gas-phase measurements suggest that $C_9H_{12}O_x$ was more than 10 times more abundant than $C_9H_{14}O_x$, as shown in Figure 1b and Figure 1d. Similar discrepancies were observed for OH-cresol and $O_3$-limonene oxidation products regarding the relative intensity of $C_7H_{6,8,10,12}O_x$ and $C_{10}H_{14,16}O_x$, respectively. These disagreements cannot be explained by gas to particle partitioning: For example, the EESI-TOF signal intensity ratios of

$C_9H_{14}O_4$-to-$C_9H_{12}O_4$ and $C_9H_{14}O_5$-to-$C_9H_{12}O_5$ are 1.6 and 10.5, respectively, while the Vocus-PTR gas-phase concentration ratios of $C_9H_{14}O_4$-to-$C_9H_{12}O_4$ and $C_9H_{14}O_5$-to-$C_9H_{12}O_5$ are 0.1 and 1.4, respectively, as shown in Figure 1b and 1d. The $C_9H_{14}O_x$ products of TMB may originate through ring-opening reactions or contain more hydroperoxide moieties (R-OOH), whereas $C_9H_{12}O_x$ products may originate through ring-retaining reactions or contain hydroxyl (R-OH), aldehyde (R=O) or ketone (R=O) moieties, all of which should make the partitioning of $C_9H_{12}O_x$ more favorable towards the particle phase relative

to $C_9H_{14}O_x$ according to group contribution theory (Pankow and Asher, 2008). Different factors may contribute to the discrepancies between Figure 1b and Figure 1d: Differences in the relative response factor of EESI-TOF and/or Vocus-PTR to these compounds (e.g. $C_9H_{12}O_x$ vs.$C_9H_{14}O_x$). For instance, a comparison of gas-phase measurements by nitrate ion atmospheric pressure chemical ionization, iodide ion chemical ionization, and Vocus-PTR shows that they differ in the reported relative abundances of OH-TMB oxidation products (Wang et al., 2020). In addition, artefacts such as –$H_2O$ loss from ion

fragmentation in the Vocus-PTR, which is not uncommon for PTR-based techniques (Yuan et al., 2017), or the formation of



[M+H$_2$O]$^+$ adducts in the EESI-TOF, which has been reported previously to occur to a minor extent (Lopez-Hilfiker et al., 2019), also contribute to the discrepancies between EESI-TOF and Vocus-PTR. These uncertainties complicate our data interpretation but do not change the overall trend in the sensitivity estimation, as shown in Figure S7. For simplicity, it is assumed that fragmentation in Vocus-PTR or adduct formation in the EESI-TOF was negligible for the remainder of the study

unless specified otherwise.



**Figure 1.** (a) Time-series of particle phase OH-TMB oxidation products, C$_9$H$_{14}$O$_x$ observed as Na$^+$ adduct ions by the EESI-

TOF and the total organic aerosol concentration as observed by the AMS during three separate seed injection events. (b) Relative abundances of C$_9$H$_{12-20}$O$_{1-9}$ in the particle phase based on the ion intensities observed by the EESI-TOF averaged over all seed injection events. (c) Time-series of select gas phase C$_9$H$_{14}$O$_x$ products observed as protonated ions by the Vocus-PTR, and the estimated condensation sink. (d) Relative abundances of C$_9$H$_{12-20}$O$_{1-9}$ in the gas phase based on the ion intensities observed by the Vocus-TOF averaged over all seed injection events. The decrease in EESI-TOF intensity in (a) at 25 min





corresponds to an automated background filter measurement. The gap in the Vocus-PTR time series in (c) at around 16 min
corresponds to an automated background zero air measurement.

In the absence of any organic vapor wall loss, the amount of condensed organic material, $P_{cond}$ in response to seed
injection is equal to the decrease in the gas-phase concentration, $\Delta Gas$. In the presence of vapor wall loss, the expected $P_{cond}$
would be greater than the observed $\Delta Gas$. In addition, for sufficiently volatile species, the relative change in the gas-phase
concentration is small. This makes quantification of $P_{cond}$ and $\Delta Gas$ based on the difference between the remaining gas-phase
concentration $G_{remain}$, from the steady-state concentration $G_{SS}$, increasingly untenable. The sensitivity of $P_{cond}$, $G_{remain}$, and $\Delta Gas$
to changes in wall loss rate, $k_w$, organic aerosol concentration, $OA$, condensation sink, $CS$, and the saturation vapor
concentration, $C^*$, are modeled using KinSim and shown in Figure 2a-c, assuming a base case condition of 0.04 s$^{-1}$ $k_w$, 20 µg
m$^{-3}$ $OA$, 1 s$^{-1}$ $CS$. As shown in Figures 2a-b for species with $C^*$ below $10^{-2}$ µg m$^{-3}$ (e.g. LVOC and ELVOC), the amount of
vapor condensed is determined primarily by the $CS$ and $k_w$, where the back reaction (i.e. evaporation from the particle phase)
is negligible. Conversely, for species with $C^*$ above $10^4$ µg m$^{-3}$ (e.g. IVOC and VOC), negligible condensation ($P_{cond}$ and $\Delta Gas$
are <1% of $G_{ss}$) can be expected. Figure 2c suggests that $P_{cond}$ and $G_{Remain}$ are expected to be linearly anti-correlated, regardless
of $C^*$, where the slope becomes steeper as $k_w$ increases. By extension, at a given CS and OA concentration, $P_{cond}$ could be
calculated from the change in gas-phase concentration, $\Delta Gas$ (i.e. $G_{SS}$ - $G_{Remain}$) by applying a uniform, $k_w$-dependent scaling
factor.

The expected behavior of $P_{cond}$ under observed $CS$ and $OA$ conditions for compounds of varying $C^*$ is modeled and
shown in Figure 2d, which suggests that it may be possible to constrain the $C^*$ of some semivolatile compounds based on $P_{cond}$
(normalized to its maximum observed value) as a function of $CS$ using only EESI-TOF data if the $G_{ss}$ is constant between seed
injections. To identify the applicable $C^*$ range, we calculated the ratio of $P_{cond}$ to $G_{SS}$ for compounds of different $C^*$ at different
$CS$ conditions as shown in Figure 2e. The inter-correlations of the normalized $P_{cond}$ for compounds of different $C^*$, similar to
those shown in Figure 2d, are shown in Figure 2f for the TMB system, where the maximum $CS$ fitted was 0.83 s$^{-1}$. For
compounds with $\log(C^*)$ lower than 1.18 or higher than 2.09, the normalized $P_{cond}$ trends are indistinguishable from those with
lower or higher $C^*$, respectively, as shown Figure 2d and Figure 2f. While possible, constraining the $C^*$ of OA components
based solely on the particle uptake is limited to a narrow $C^*$ range under our experimental conditions. Increasing the $CS$ and/or
OA concentration could extend the constrainable $C^*$ range, albeit at the risk of primary ion suppression under high loading
conditions. See Supplementary Information Section 5 and Figure S4 for further discussions.





**Figure 2.** Modeled trend of the remaining gas-phase concentration, $G_{Remain}$ and the condensed vapor concentration $P_{cond}$ relative to the steady-state gas-phase concentration prior to seed injection, $G_{ss}$ at various organic aerosol (OA) concentrations, condensation sinks (CS), or assumption of the wall loss rates, $k_w$ for oxidation products of varying saturation vapor concentrations, $C^*$. (a) shows the ratio of $G_{Remain}$ to $G_{ss}$ as a function of $\log(C^*)$ for the base case scenario of 20 µg m$^{-3}$ OA, 1 s$^{-1}$ CS, and 0.04 s$^{-1}$ $k_w$, as well as alternative scenarios. Compounds with $\log(C^*) > 5$ and $< -2$ were modeled but not shown, as their trends are similar to that of compounds with $\log(C^*) = 5$ and -2, respectively. Similarly, (b) shows the ratio of $P_{cond}$ to $G_{ss}$





as a function of log($C^*$) under various combinations of OA, $CS$ and $k_w$. (c) shows the correlation of $P_{cond} / G_{ss}$ and $G_{remain} / G_{ss}$, which is similar regardless of the $C^*$, for different $k_w$ values. (d) shows the trend of $P_{cond,}$ normalized to its maximum, for

products of varying $C^*$ as a function of $CS$ for $k_w$ = 0.04 s$^{-1}$. Compounds with log($C^*$) > 4 and < -1 were modeled but not shown, as their trends are similar to that of compounds with log($C^*$) =4 and -1, respectively. Observed $CS$ and OA from the TMB experiment are used to simulate the uptake trend shown in (d), as opposed to the hypothetical conditions used for simulations shown in (a-c). (e) shows the modeled ratio of $P_{cond}$ to $G_{SS}$ for compounds of varying $C^*$ under different $CS$ and OA, which correlated roughly linearly with CS, conditions. (f) The expected inter-correlation of the normalized $P_{cond}$ trends for compounds

of varying log($C^*$) as a function of $CS$ for the TMB system, e.g. $R^2$ of pairwise linear regressions for traces shown in (d) or vertical slices in (e). Regions with $R^2$ above 0.99 are considered as unreliable for constraining $C^*$ based solely on normalized $P_{cond}$. The ratios of $G_{remain}$ and $P_{cond}$ to $G_{SS}$, which are independent of the production rate, are used as dimensionless quantities instead of their absolute values for ease of representation.

The near-molecular EESI-TOF sensitivities, $RF^*_x$ in ions$^{-1}$ ppb$^{-1}$ are calculated as the slope of the linear regression of [M+Na]$^+$ intensity (in ions s$^{-1}$) observed by the EESI-TOF as a function of $\Delta Gas$ (in ppb) determined by the Vocus-PTR for the corresponding species, i.e. [M+H]$^+$. Based on the KinSim model results, we restricted the sensitivity analysis to compounds with expected $\Delta Gas \geq 1\%$ of $G_{SS}$ under the highest observed OA and $CS$ conditions, which correspond to compounds with log($C^*$) < ~3.6. Assuming that the $C^*$ estimated using the molecular corridor method (Li et al., 2016) has at least one order of

magnitude uncertainty (i.e. ±1 in the calculated log($C^*$)), we relax the maximum log($C^*$) threshold to 4.6. No correction for vapor wall loss was applied, which should only affect the absolute but not the relative $RF^*_x$ for a given $k_w$ (Figure 2c). As shown in Figure 3a, the estimated $RF^*_x$ ranges from $10^3$ to $10^6$ ions s$^{-1}$ ppb$^{-1}$, with the majority falling within 1 order of magnitude of their geometric mean at ~$10^{4.5}$ ions s$^{-1}$ ppb$^{-1}$. For OH-cresol and OH-TMB oxidation products, the $RF^*_x$ exhibits positive correlation with molecular weight and oxygen content, as shown in Figure S8, which is consistent with previous

findings on the positive correlation of the Na$^+$ adduct ionization efficiency with molecular volume and ion-dipole interactions during ESI (Oss et al., 2010; Mayhew et al., 2020; Kruve et al., 2013). Similar correlations were not observed for O$_3$-limonene oxidation products, which are likely due to differences in their molecular structures. It has been shown for isomers that their respective [M+Na]$^+$ adducts can assume different conformations with different binding energies (Yang et al., 2017; Bol et al., 2017). Figure 3b compares the estimated EESI-TOF sensitivity of analytes from TMB oxidation with the ones from cresol or

limonene oxidation. The sensitivity of isomers, comprised exclusively of C$_{8-10}$ compounds, mostly agree within a factor of 20. The isomers derived from creosol have lower sensitivities than those derived from TMB, likely due to the cresol oxidation producing more (~75%) ring-retaining products (Schwantes et al., 2017), making them less polar than the ring-opening isomers, which may limit its extraction efficiency or the stability of the Na$^+$ adduct. A similar argument could be made for the increased isomer sensitivity of limonene oxidation products, e.g. C$_9$H$_{12}$O$_4$, which would likely be a ring-retaining product of

TMB (C$_9$H$_{12}$), but a ring-opening product of limonene (C$_{10}$H$_{16}$). Alternatively, the $RF^*_x$ of C$_9$H$_{12}$O$_x$ may be underestimated for the TMB oxidation products due to H$_2$O loss from C$_9$H$_{14}$O$_x$ in the Vocus-PTR, which overestimates the $\Delta Gas$ and $P_{cond}$ of





$C_9H_{12}O_x$. Under the assumption that all ions undergo $H_2O$ loss inside the Vocus-PTR, closer agreements (within a factor of 5) in $RF^*_x$ were observed for isomers between the TMB and limonene systems, as shown in Figure S7d.

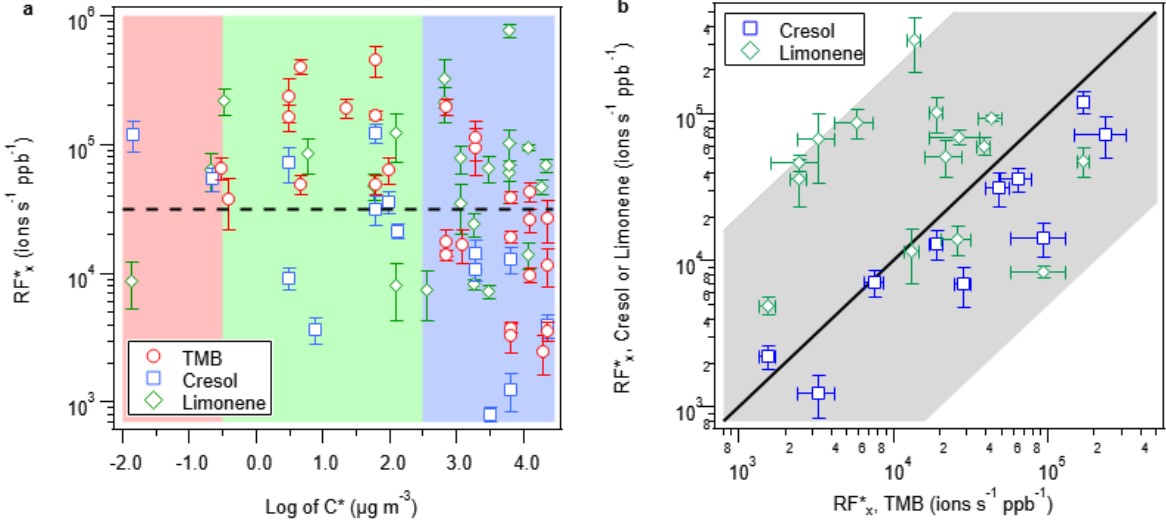


**Figure 3.** (a) Compound-dependent EESI-TOF response factor, $RF_x^*$ of OH-TMB, OH-cresol, or $O_3$-limonene oxidation products calculated as the linear regression slope of the observed rise in the particle-phase EESI-TOF ion intensity and the corresponding decrease in the gas-phase Vocus-PTR mixing ratio. From each reaction system, only oxidation products with $R^2 \geq 0.5$ for the linear regression of EESI-TOF signal increase and Vocus-PTR mixing ratio decreases are shown. The average log($RF^*_x$), ~4.5 is shown as the dashed line. Red, green, and blue shadings indicate the log($C^*$) range for low-volatility, semi-volatile, and intermediate volatility organic compounds, respectively (b) Comparison of the $RF^*_x$ for analytes with identical molecular formula observed in different systems. The 1-to-1 line is shown in solid black. The shaded region represents a factor of 20 deviation from the 1-to-1 line. The error bars shown correspond to the uncertainty of the fitted slopes.

### 3.3 Regression analysis

As discussed above, the near-molecular EESI-TOF sensitivity can be highly variable. While it is possible to estimate the binding energy (and by proxy the $RF^*_x$) of $[M + Na]^+$ with an assumed adduct conformation via quantum chemical calculation, it would require a priori knowledge of the analyte molecular structure, which is not readily obtainable. Given the elemental composition which is obtainable with high-resolution mass spectrometry, i.e. $n_C$, $n_H$, and $n_O$, we show in Figure 4a that it is possible to predict the EESI-TOF $RF^*_x$ within a factor of 5 using either linear regression or non-parametric regression

models for OH-TMB oxidation products. Feature selection results shown in Figure S8 and Table S2 suggest that, despite their differences, all regression models evaluate $NI_{CO}$ as one of the most important features in predicting the EESI-TOF $RF^*_x$ for OH-TMB oxidation products. The positive correlation between $RF^*_x$ and $NI_{CO}$ is intuitive, as compounds with higher $NI_{CO}$ values also tend to be larger (i.e. higher $n_C$ and molecular volume) and/or more functionalized (i.e. higher $n_o$), possibly



enhancing the chelation of analyte M with Na⁺ and the binding energy of the [M+Na]⁺. In addition, higher $NI_{CO}$ roughly

translates to a lower $C^*$, and therefore likely a higher Henry's law constant (Hodzic et al., 2014), i.e. higher solubility and liquid-liquid extraction efficiency. The better performance obtained by nonparametric models compared to linear models, and the importance of interaction terms such as $NI_{CO}$ indicate that the $RF^*_x$ is not a linear function of the elemental composition. It is also important to recall that fragmentation reactions such as dehydration of carboxylic acids occur in PTR; for similarly functionalized compounds, the degree of fragmentation increases with the size of the molecule (Yuan et al., 2017). When not

corrected for, the fragmentation reactions would cause (Vocus-)PTR to underestimate the concentration of larger, more oxidized compounds. This would lead to overestimation of EESI $RF_x^*$ for larger compounds and vice versa, partially accounting for the observed trend in the EESI $RF_x^*$ with $NI_{CO}$.

Although the observed $RF_x^*$ are specific to our instrumental and ES conditions, similar $NI_{CO}$ sensitivity dependences may hold for other EESI(-TOF) systems, at least for TMB oxidation products. Regression models may perform very well for

the training set (see Figure S11) and may even allow for qualitative prediction for a similar dataset, as shown in Figure S12a for the prediction of $RF^*_x$ of cresol oxidation products by the model trained on TMB oxidation products. However, the regression models may not extrapolate well to a more diverse (Figure S10) or distinct system, as shown in Figure S12b for the prediction of $RF^*_x$ of limonene oxidation products using the regression model trained on TMB oxidation products. As discussed in Section S7, the inclusion of $RF^*_x$ estimated for OH-cresol and $O_3$-limonene oxidation products in the regression analysis

significantly degrade the prediction accuracies of all regressors. Without any knowledge of the VOC precursor identity, the regression models did not perform better or performed only marginally better than simply assuming a uniform $RF^*_x$ equal to that of the geometric mean, ~$10^{4.5}$ ions s⁻¹ ppb⁻¹, as shown in Figure S10b. In this case, all regression models identified aromaticity ($X_C$) as an important feature, which can be thought of as an attempt by the models to learn the identity of the VOC precursor or the structure of the oxidation products that derive from said precursor. If the VOC precursor can be constrained,

whether explicitly or perhaps using properties such as $X_C$ as surrogates, it is possible to obtain reasonable predictions (e.g. with a factor of 5 of the measured value) for $RF^*_x$ using nonparametric models such as GBR, as shown in Figure 4b and Figure S10c, which suggests $n_O$ and mass defect (related to the molecular weight and oxygen content) as some of the important predictors for $RF^*_x$, along with the VOC precursor identity. Additional structural information obtained using, for example, ion mobility spectrometry or tandem mass spectrometry would likely further improve $RF_x^*$ prediction in lieu of prior knowledge

of the VOC precursor when used in parallel or in tandem with EESI-TOF.



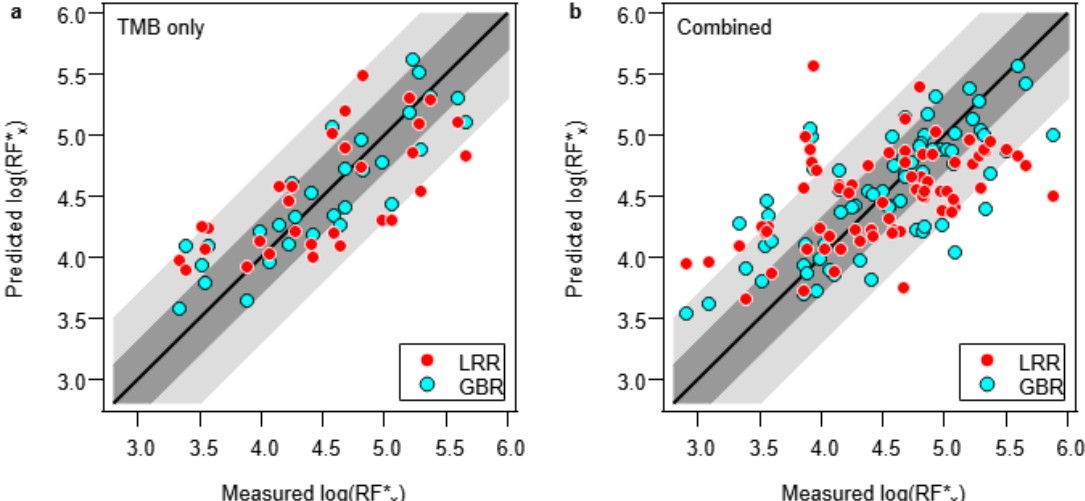

**Figure 4.** Comparison of predicted and measured log of $RF^*_x$ value using linear ridge regression (LRR) and gradient boosting regression (GBR) for (a) the TMB system only, or for (b) all three VOC systems combined, where the VOC precursor identity was digitized and included as one of the features. The 1-to-1 line is shown in solid black. The lighter and darker shaded regions represent a factor of 2 and 5 deviations from the 1-to-1 line, respectively. Model accuracies (see Eq. S21) are approximately 0.69 for GRB and 0.49 for LRR in (a), and 0.49 for GBR and 0.24 for LRR in (b). See Section 7 and Figure S10 in the Supplementary Information for details on feature selection and model validation.

### 3.4 Bulk sensitivity

The sum of the background-corrected $[M+Na]^+$ ion intensities, weighted by the molecular weight of individual analytes, correlates linearly with the bulk organic aerosol concentration measured with the AMS, as shown in Figure 5a. The bulk relative response factor $RRF_x$ normalized to that of OH-TMB oxidation products, is approximately 0.23 and 0.54 for SOA produced from OH-cresol and $O_3$-limonene reactions, respectively. The bulk $RRF_x$ observed in this study using 1:1 acetonitrile:water ES solvent is consistent with the previous study using 1:1 methanol:water as ES solvent, where the EESI-TOF bulk $RRF_x$ towards OH-oxidation products of TMB was shown to be approximately 1.8 and 5 times higher than that of OH-oxidation products of toluene and α-pinene, respectively (Lopez-Hilfiker et al., 2019). Note that $o$-cresol is one of the main first generation oxidation products of toluene, whereas limonene is a structural isomer of α-pinene. The total OA mass concentrations calculated using the predicted near-molecular response factor, $RF^*_x$, from EESI-TOF ion intensities overestimate the bulk OA concentration by up to a factor of 4 but otherwise agree with the OA concentration measured by AMS, as shown in Figure S13.

In addition to $[M+Na]^+$, a substantial amount of nitrogen(N)-containing ions was observed. Because no nitrogen oxides ($NO_x$) or reduced nitrogen species were injected into the OFR, these N-containing ions are assumed to correspond to $[M+C_2H_3NNa]^+$, if they contain at least two carbon and three hydrogen atoms. It is possible that $NH_4NO_3$ decomposition may



serve as a $NO_x$ source during seed injection, but this effect is likely negligible given the highly oxidizing environment in the OFR (i.e. > 10 ppm $O_3$). The average ratio of $\sum[M+C_2H_3N+Na]^+$ to $\sum[M+Na]^+$ ranges from 0.18 for TMB-OH SOA to 0.32

for limonene-$O_3$ SOA as shown in Figure 5b, and may have contributed to the discrepancies in $RF_x$, calculated for $[M+Na]^+$ adducts. The $[M+C_2H_3N+Na]^+$-to-$[M+Na]^+$ ratio also appears to be species-dependent, which likely reflects the differing $[M+C_2H_3N+Na]^+$ adduct stability, which is beyond the scope of this study. Caution should be taken in ion assignment, especially when nitrogenated organic aerosol components are expected. One solution is to use pure water or a mixture of water with labeled $C_2H_3{}^{15}N$ (at the cost of increased spectral complexity) as the ES solution. Alternatively, one could increase the

collision-induced dissociation energy to decluster the $[M+C_2H_3N+Na]^+$ adduct ions, which seem to have lower binding energies than $[M+Na]^+$ adducts (Lee et al., 2020).

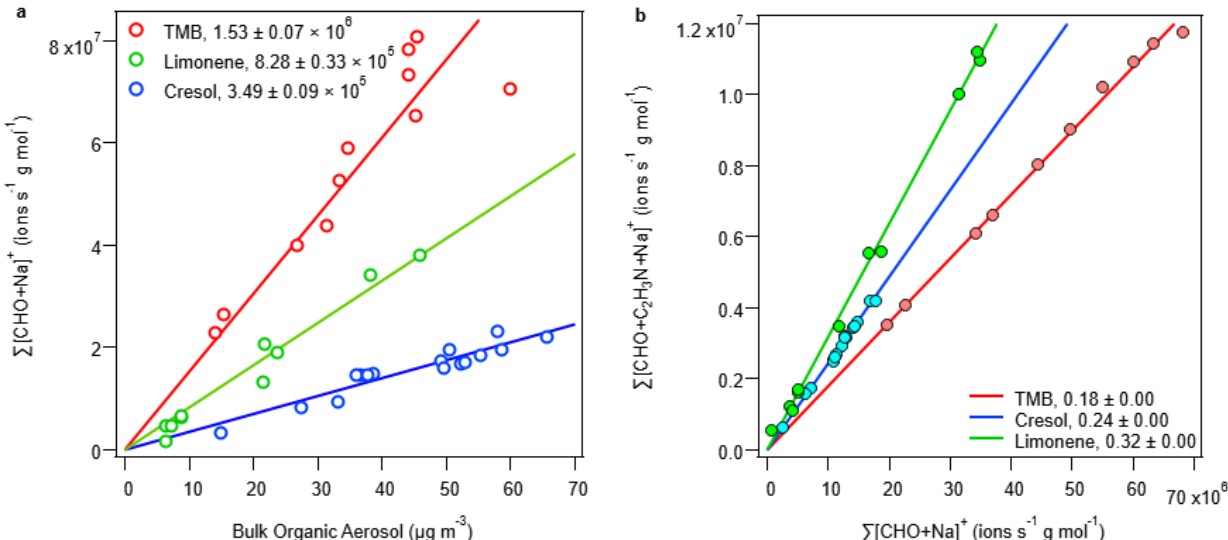

**Figure 5.** Correlation of (a) the total EESI-TOF $[M+Na]^+$ ion intensities with the bulk organic aerosol concentration as

determined by the AMS and (b) the sum of background-subtracted, MW-weighted EESI-TOF $[M+C_2H_3N+Na]^+$ and $[M+Na]^+$ signals observed during the seed injection periods, excluding any analytes observed in only one of the two forms, i.e. $[M+C_2H_3N+Na]^+$ or $[M+Na]^+$. Linear ordinary least square regression with forced 0 intercept is applied to determine the bulk EESI-TOF response factor in (a) and the average ratio of $[M+C_2H_3N+Na]^+$ to $[M+Na]^+$ in (b). For each species M, its EESI-TOF ion intensity measurement is weighted by its molecular weight in order to compare ion flux measurement by EESI-TOF

with mass measurements by AMS.

**4. Conclusion**

In this study, we conducted organic aerosol uptake experiments in an oxidation flow reactor to constrain the EESI-TOF response factor to biogenic and anthropogenic secondary organic aerosols. Intercomparison with the AMS shows that the





total EESI-TOF signal responds quantitatively to the bulk organic aerosol concentration even with significant concentrations

of inorganic aerosol present (up to 7 mg m$^{-3}$). The relative bulk response factor (i.e. ratio of summed EESI signal, weighted by molecular weight, to AMS OA measurement) is shown to be highest for the oxidation products of 1,3,5-trimethylbezene (1.00), followed by $d$-limonene (0.54) and $o$-cresol (0.23), consistent with previous results (Lopez-Hilfiker et al., 2019). Quantification of gas-phase oxidation products by a Vocus-PTR at steady-state and during organic aerosol uptake allowed us to constrain the EESI-TOF sensitivity for a range of semi-volatile organic compounds. The measured near-molecular response

factor, $RF^*_x$, ranges from $10^3$ to $10^6$ ions s$^{-1}$ ppb$^{-1}$, though mostly fall within $\pm 1$ order of magnitude of $\sim 10^{4.5}$ ions s$^{-1}$ ppb$^{-1}$. Isomer sensitivities were shown to vary by up to a factor of 20, with those showing more aromatic characteristics having lower sensitivities. Based on the measured elemental composition and properties derived from it, regression models are shown to be able to predict the measured sensitivity within a factor of $\pm 5$ for OH-TMB oxidation products. The regression models trained with the TMB dataset can also produce qualitative predictions of $RF^*_x$ for oxidation products derived from a structurally similar

VOC precursor (i.e., $o$-cresol), though not for those derived from a distinct VOC precursor (i.e. $d$-limonene). Regression analyses suggest that carbon-oxygen non-ideality ($NI_{CO}$, Eq. 7) is one of the key predictors for EESI-TOF response factor to OH-TMB oxidation products, which points to enhanced ability of an organic compound to chelate with Na$^+$ as it increases in size (e.g. number of carbon) and/or functionalization (e.g. number of oxygen) as the reason for the sensitivity increase. Increases in $NI_{CO}$ can also manifest as decreases in volatility, which may enhance solubility and liquid-liquid extraction

efficiency during EESI. For a mixed VOC system, knowledge of the SOA precursor identities can help to constrain the EESI-TOF response factors. The total OA concentration estimated using the EESI-TOF measurements and the predicted $RF^*_x$ values show reasonable agreement with the observations by the AMS and SMPS within a factor of 4. If the SOA precursor is unknown, the $RF^*_x$ prediction by regression models are marginally better than assuming a uniform EESI-TOF sensitivity. For bulk concentration estimation and time-series analysis, uniform sensitivity is a reasonable assumption, as seen in good agreements

between EESI-TOF and AMS measurements for ambient organic aerosol samples despite variabilities in individual $RF^*_x$ (Liu et al., 2019; Qi et al., 2019; Stefenelli et al., 2019), likely due to the law of averages. For estimating the relative contribution by individual species, the parameterization of $RF_x^*$ developed in this study may be applicable for SOA sources dominated by aromatics chemistry.

To our knowledge, this is the first study to constrain the EESI-TOF response factor to organic aerosol components

without the use of chemical standards, many of which cannot be purchased commercially or synthesized. We show that it is possible to semi-quantitatively resolve the organic aerosol composition at the near-molecular level with high time resolution. One limitation of the study is that the analytes with overlapping coverage in the EESI-TOF and Vocus-PTR do not include any low-volatility compounds, extremely low-volatility compounds, or organic nitrates. Future studies should consider extending the oxidation timescale, minimizing wall loss, and utilizing techniques more suited for the quantification of

moderately to highly oxygenated compounds at low concentrations, such as a series of chemical ionization mass spectrometry. Future studies should also deconvolve the effects of liquid-liquid extraction efficiency, ionization efficiency, and ion



transmission efficiencies which all contribute to the overall EESI sensitivity observed. Online structural elucidation would also be instrumental in the identification and quantification of organic aerosol components.

*Data Availability*. The data presented in the text and figures will be available at the Zenodo Online repository (https://zenodo.org) upon final publication. Underlying data are available upon request.

*Author contributions.* DSW, CPL, JD, JGS, and DMB designed the experiment. DSW, CPL, JEK, MRC, FM, YT, and DMB performed the experiments and collected the data. DSW, JEK, CPL, YT, and DMB analyzed the data. DSW, CPL, JEK, UB,
IEH, JGS, and DMB prepared the manuscript. All authors contributed to the data interpretation and manuscript revision.

*Competing interests.* JEK, MRC, and FM are employed by Aerodyne Research Inc., which commercializes the Vocus-PTR, the EESI-TOF, and the AMS. The other authors declare no conflict of interest.

*Acknowledgement.* We thank Houssni Lamkaddam and Mao Xiao for their helpful discussions.

*Financial support.* This project has received funding from the European Union's Horizon 2020 Research and Innovation Program under the Marie Skłodowska-Curie grant agreement no. 701647 and through the EUROCHAMP-2020 Infrastructure Activity under the grant agreement no. 730997, as well as from the Swiss National Science Foundation (20020_172602,
BSSGI0_155846).

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
