# Peer review of "Constraining the response factors of an extractive electrospray ionization mass spectrometer for near-molecular aerosol speciation"

_Atmospheric Measurement Techniques, 2021_

## Author Comment (AC1)

**Author responses to referee comments**

We thank the referees for their comments and suggestions. Our point-by-point responses are below in normal font. Original comments by the referees are reprinted in **bold**. Text changes are highlighted in **blue**.

**Referee 1:**

This manuscript describes an effort to estimate the sensitivity of an extractive electrospray ionization TOF-MS to multifunctional organic compounds produced from the oxidation of 1,3,5-trimethylbenzene, d-limonene and o-cresol. The authors use the semi-quantitative response of a Vocus-PTR instrument to the same compounds (or molecular formulas) measured in the gas phase to estimate the sensitivity of the EESI-ToF-MS to those molecular formulas in the particle phase. The authors find a very wide range in EESI response factors, largely driven by molecular size/volume and degree of oxidation or functionalization. Further, comparison of summed EESI signals with total organic content from an aerosol mass spectrometer indicates promise for bulk quantification.

This is an interesting analysis that I believe should be published in AMT, given that the following comments and suggestions can be adequately addressed.

**Major Comments:**

(1) As the authors state in their abstract, these results demonstrate that some level of quantification with the EESI-ToF-MS is possible without standards; however, this analysis includes a series of logical leaps that lead to significant uncertainty in resulting EESI response factors. The authors state that the Vocus-PTR -based EESI response factors range from 10^3 to 10^6 ions/s/ppb, with an geometric mean of 10^4.5 ions/s/ppb. For a given EESI RF for a specific compound, what is the associated uncertainty? (e.g., determined by propagating the uncertainty in calculated k\_MH applied to Vocus data, described in L125-127 to become "more uncertain" as parameterizations are applied to more oxygenated species, through to the resultant EESI response factor). Related to this, how are the error bars in Figure 3a generated?

In the AMTD version, error bars in Figure 3a are the standard deviation of the slope of EESI signal (in ions s-1) as a function of Vocus measurement (in ppb) obtained using ordinary least square (OLS) fitting, without taking into account the uncertainties of either the explanatory variable or responses themselves. For the revised manuscript, we performed orthogonal distance regression (ODR) of EESI vs. Vocus measurements, taking into account their respective measurement uncertainties, including that of  $k_{MH}$  as reported by Sekimoto et al. (2017), i.e. 50%. Error bars in Figure 3 are updated and they reflect the standard deviation of the fitted slope, i.e. the EESI  $RF_x^*$ .

Results obtained by OLS and ODR agree within a factor 2, with ODR results having lower standard deviations. A new figure (Figure S7) comparing the OLS and ODR results (shown below) is added to the S.I. Related to this, we have updated our regression analysis using ODR results.

Further clarification for  $k_{MH}$  is provided below in response to comment 2.

**Figure S7**. Comparison of the response factor  $(RF_x^*)$  values determined using ordinary least square (OLS) and orthogonal distance regression (ODR). Uncertainties in the explanatory and response variables are taken into consideration by ODR during fitting. Vertical and horizontal error bars shown represent the standard deviation of the fitted slope of EESI-TOF vs. Vocus-PTR measurements, i.e.  $RF_x^*$ , obtained using ODR and OLS, respectively.

Line 198: "The  $RF_x^*$  value corresponds to the slope of  $I_x$  as a linear function of  $P_{cond,x}$ . Similar results for  $RF_x^*$  were obtained using ordinary least square (OLS) and orthogonal distance regression (ODR), as shown in Figure S7. In order to propagate various uncertainties, including that of Vocus-PTR calibration factors, we report the  $RF_x^*$  values obtained using ODR in subsequent analyses."

(2) Estimation of EESI response factors for specific molecular formulas in this work hinges on quantification of Vocus signals using k\_MH calculated according to Sekimoto et al., 2017 (in addition to assumptions about gas-particle partitioning). Under the fIMR conditions given in L 135-136 (i.e., 59 Td), is Vocus sensitivity linear with k\_MH for compounds that can be calibrated? At 59 Td, water clusters should dominate the reagent ion distribution over  $H_3O^+$ , potentially producing a regime where sensitivity is not linear with k\_MH. The authors should show that their sensitivity is linear with k\_MH (as an SI figure), thus demonstrating that they can apply k\_MH for these unknown compounds. If the Vocus sensitivity is not linear with k\_MH under these conditions, the authors should demonstrate how this impacts their use of Vocus signals to constrain EESI response.

We have added an S.I. figure (shown below) showing the correlation between the measured sensitivity and estimated  $k_{\text{MH}}$  values for the calibrants used. The sensitivities strongly correlate with  $k_{\text{MH}}$ . Assuming an intercept of 0 potentially underestimates the sensitivity of oxygenated compounds, though it is within the 50% error, which we have now incorporated in our error propagation. We have added some clarification to the main text.

Line 128: "The calculated  $k_{MH}$  values correlate linearly with the measured sensitivities for the calibrants used, as shown in Figure S2b."

Line 133: "A detailed characterization of  $k_{MH}$  and  $F_{MH}$  is beyond the scope of the current study. Instead, we assume a 50% uncertainty for the estimated Vocus-PTR sensitivity as reported by Sekimoto et al. (2017) and propagate this uncertainty in subsequent analyses."

Figure S2b. Measured sensitivity as a function of the kinetic capture rate,  $k_{MH}$  for compounds in a multicomponent calibration tank. The linear regression line with forced 0 intercept was used to estimate sensitivities for additional uncalibrated compounds.

(3) A statement prominent in the conclusions section is that the EESI-ToF-MS responds quantitatively to organic aerosol concentrations even with significant inorganic aerosol present. This seems a rather important conclusion of this paper, but the exact origin of this conclusion is somewhat unclear. Please make explcit whether this conclusions arises directly from the linear relationship between summed EESI signals and AMS OA, or some other specific result.

This statement is based on the linear relationship between the summed EESI signals and the AMS as shown in Figure 5. The concentration range of inorganic seed aerosol used is shown in Figure S3. We have revised this sentence in the Conclusion to make the connection more explicit.

Line 478: "Intercomparison with the AMS shows that the total EESI-TOF signal responds quantitatively to the bulk organic aerosol concentration even with significant concentrations of inorganic aerosol present (up to 4.7 mg m-3 of NH4NO3 seed particles, Figure S3), based on the linear relationship between the summed EESI-TOF signals of the Na+ adduct ions and the AMS measurements of the bulk OA concentrations (Figure 5)."

**(4) This manuscript contains a large number of symbols and acronyms. A glossary of terms might be useful as an appendix to this paper so the reader can reference it easily.**

We have added a glossary as Appendix. A of selected symbols and acronyms, as suggested.

**Specific Comments:**

**L21: Inorganic coating or inorganic seed particles?**

It should be inorganic seed particles. We have corrected the text:

Line 21: "The EESI response factors towards bulk SOA coating and the inorganic seed particle core were constrained by intercomparison with the AMS."

**L48: The authors may also wish to cite https://pubs.acs.org/doi/10.1021/acsearthspacechem.9b00312 in this context**

We have added this reference on thermal desorption artefacts.

**L70: What additional differences can be contributed by "operators" beyond the instrument parameters already listed?**

We intended to use "operators" as a catch-all term for variations in experimental conditions that would too numerous to list, such as temperature, humidity, pressure drops etc.. We have modified the text for clarity.

Line 68: "However, the relative ionization efficiencies (RIE) of EESI or ESI (where the analyte is directly infused into the ES solution) towards different compounds could vary by orders of magnitude depending on the instrument setting, sample matrices, electrospray solutions, and other experimental conditions."

**L85-101: Please state clearly whether the seed aerosol was polydisperse or monodisperse**

The seed aerosol was polydisperse. We have added the following clarification:

Line 102: The particle count mean diameter of the polydisperse seed aerosol ranged from 60 to 100 nm during seed injections.

**L180-185 (an in conclusions): Can these different factors contributing to EESI RF\_x be separated? Some discussion on how these could be separated, if at all, would be useful here.**

We have added perspectives on potential strategies and challenges in separating some factors in the Conclusion section.

Line 513: "Future studies should also deconvolve the effects of liquid-liquid extraction efficiency, ionization efficiency, and ion transmission efficiencies which all contribute to the overall EESI sensitivity observed. Isolating each contributing factor is challenging. For instance, the ion transmission efficiency of the mass analyzer may be characterized by splitting the analyte ion flow between the MS inlet and a Faraday cup electrometer to monitor the incoming flux (Heinritzi et al., 2016), although changes made to the ionization region geometry or flow may alter ES characteristics. One could also potentially probe the extraction efficiency by comparing EESI and infusion ESI results, although the two closely related techniques may differ in their ionization efficiencies and matrix effects."

**L390-391: Both larger molecules/higher molar volume and higher solubility in water are somewhat opposing features. Is it clear from this analysis which features are dominant?**

The study by Hodzic et. al. (2014), referenced here, reported a negative correlation between saturation vapor pressure and Henry's law constant, where less volatile SOA components were in general more soluble in water. This suggests that the (decreases in) volatility is, in general, more dominant than (increases in) molar volume in determining the solubility of SOA components.

**L481: "law of averages" -- please be more specific**

Comparing the sum of EESI-TOF signals vs. AMS measurements, the bulk ("average") EESI-TOF sensitivity varies much less ( $<< \pm 1.5$  orders of magnitude) over the course of an ambient campaign compared to the range of individual  $RF^*_x$ . We have rephrased the text as follows,

Line 500: "For bulk concentration estimation and time-series analysis, uniform sensitivity is a reasonable assumption, as seen in good agreements between EESI-TOF and AMS measurements for ambient organic aerosol samples (Qi et al., 2019; Stefenelli et al., 2019),

where the bulk response factor remains consistent (in time) despite variabilities in individual  $RF_{x}^{*}$ ."

**Referee 2 (Dr. Demetrios Pagonis):**

This manuscript describes the use of PTR-MS and AMS to constrain the response factors of EESI-TOF. The authors oxidized VOCs in a flow tube, varied the seed aerosol concentrations (and thereby the condensational sink), and compared the reduction in gasphase concentrations of organic compounds measured by Vocus PTR-MS to the increase in aerosol-phase signal measured by EESI-TOF. This dataset provides the most comprehensive attempt to constrain EESI-TOF response factors to date and will be valuable to the continued development of EESI-TOF as a technique for studying atmospheric aerosol. I recommend that this manuscript be published in AMT after the following comments are addressed:

1) I agree completely with Referee #1's comment that the authors must expand discussion of how the uncertainty in the Sekimoto et al. 2017 parameterization propagates through to the calculated  $RF_x^*$  in this work. The error bars from the fit uncertainty shown in Fig. 3 suggest to me that the authors' methods produced good fits between the EESI-TOF and Vocus signals, and so the 50% uncertainty from the PTR parameterization is going to be a major contributor to the overall uncertainty.

We have revised our analysis using orthogonal distance regression (ODR) to incorporate the uncertainty of  $k_{MH}$  (50%) into our error propagation, as detailed above in our response to Referee 1's comment on this subject.

2) The authors' assumption that there is no fragmentation in the Vocus PTR-MS does not seem consistent with the literature on PTR fragmentation, especially for monoterpenes. See for example the recent GC-Vocus work from Claflin et al. 2021 AMT which documents limonene fragmentation of 50% (Table 3). While the extent of fragmentation will vary significantly with drift tube conditions, the oxidation products of limonene (e.g. alcohols, peroxides) typically show fragmentation of 70% or more (Pagonis et al. 2019). I am a coauthor on both of these papers and so I don't insist that the authors cite either one. In my opinion, fully constraining the fragmentation patterns of the compounds in these oxidation systems is far beyond the scope of this work, so my suggestion for the manuscript is to move Fig. S7 into the main text. This analysis nicely shows the extent to which fragmentation could impact the results, and should be featured prominently since the true relationship is likely somewhere between Fig. 3 and Fig. S7.

We have replaced Fig.3 with Fig. S7 and added additional discussion on the impact of fragmentation, including references to the two studies mentioned here.

**Line 113:** "The proton-transfer-reaction is exothermic and the ionization-induced fragmentation occurs to varying degrees (~0 to ~100 %) as the result of dehydration, H2 elimination, alkyl group loss, or HNO3 loss (**Claflin et al., 2021**; Duncianu et al., 2017; Leglise et al., 2019; **Pagonis et al., 2019**; Yuan et al., 2017), which can be represented by the term,  $F_{MH}^+$ ."

Line 289: "In addition, artefacts such as ion fragmentation are known to occur for PTR-based techniques (Yuan et al., 2017), which could be substantial (>70%) for limonene oxidation products (Claflin et al., 2021; Pagonis et al., 2019); the formation of  $[M+H_2O]^+$  adducts in the EESI-TOF, which has been reported previously to occur to a minor extent (Lopez-Hilfiker et al., 2019), may also contribute to the discrepancies between EESI-TOF and Vocus-PTR."

**Line 294: "A detailed characterization of the extent of ionization-related artefacts is beyond the scope of this study."**

3) The difference in predictive capability within a VOC system (Fig. 3a: TMB-only) and across the three VOCs studied (Fig S10b: all precursors, no label) gives the best chemical insight into the chemistry underlying EESI-TOF response factors. The authors have some discussion about this, but I think that this result supports further discussion. Molecular formula becomes a more useful parameterization when the carbon backbone and positions of functionalization of the analytes are consistent (a constraint brought on by using a single VOC precursor). To me, this indicates that a structure-activity relationship could be a very promising approach as e.g. EESI-MS-MS techniques are able to give information about the identity and position of functional groups.

We agree that the differences between VOC systems, specifically the impact of the VOC precursor structure, can offer structural insights into oxidation products. However, considering that several isomers (with differing sensitivities) may derive from a single VOC precursor (Bi et al., 2021), as well as the various experimental and analytical uncertainties, we prefer to be conservative with our interpretations.

**4) The EESI-TOF ion transmission as a function of m/z is left unconstrained in this work. This is a gap that I would like to see addressed more in the text, especially since it gets a full treatment on the Vocus half of the analysis.**

We have added a discussion on impact of mass transmission efficiency in Section 3.2, and on the characterization of ion transmission efficiency in Section 4. Due to possibilities of declustering of  $[M+Na]^+$  adducts, reagent ion depletion method proposed by Henritzi et. al. (2016) is not readily adaptable here, and only the Faraday cup electrometer method is mentioned in the Conclusion.

Line 412: "Furthermore, the positive dependence of  $RF_x^*$  on  $NI_{CO}$  or  $n_0$  may also reflect the impact of EESI-TOF mass transmission efficiency, where the ion optics were configured to favor the transmission of medium-to-high masses, e.g. ions with higher carbon and/or oxygen contents."

Line 515: "For instance, the ion transmission efficiency of the mass analyzer may be characterized by splitting the analyte ion flow between the MS inlet and a Faraday cup electrometer to monitor the incoming flux (Heinritzi et al., 2016), although changes made to the ionization region geometry or flow may alter ES characteristics."

**Line 89: "Milliliters" is missing a number**

We have added the approximate volume used.

**Line 90:** "Approximately 1-2 milliliters of limonene (Sigma-Aldrich, 97%), o-cresol ("cresol", Sigma-Aldrich,  $\geq$ 99%), or 1,3,5-trimethylbenzene ("TMB", Sigma-Aldrich, 98%) were placed inside a glass vial connected orthogonally to a dry clean air carrier flow to supply VOC precursors."

**Line 104: L-ToF is not informative for readers who do not use Tofwerk instruments, please replace with a description of the mass resolution for the Vocus during this study.**

We have added the mass resolving power for LTOF to the description of Vocus-PTR.

Line 104: "VOC precursors and oxidation products were measured using a Vocus-PTR long time-of-flight mass spectrometer (L-TOF), which has a mass resolving power  $(m/\Delta m)$  of approximately 8,000 at m/z 200."

**Line 108: Please confirm for the reader that the $H_3O^+$ depletion was negligible in this study.**

We have added confirmation that  $H_3O^+$  depletion was negligible.

Line 111: "If H3O+ depletion is negligible, as is the case in this study, the abundance ..."

**Line 232: Equation 7 is not for NICO**

We have added the formulation for NIco in the parenthesis.

Line 243: "...such as the carbon-oxygen non-ideality,  $NI_{CO}$  (the  $n_C n_O/(n_C+n_O)$  term in Eq.7), double bond equivalents (*DBE*, Eq. S16), and aromaticity ( $X_c$ , Eq. S17)."

Line 275: I do not follow the reasoning behind why the H12 products should favor the particle phase to a greater extent than the H14 products. I'm assuming that you're comparing SIMPOL coefficients of (hydroperoxide) against (aromatic ring + alcohol + ketone), but this is certainly not going to be clear to many readers. More broadly, I think this is overstating the precision of SIMPOL, and that it would be sufficient to rely on the signal ratios and the identical numbers of oxygen and carbon atoms to rule out differences in partitioning.

We were indeed referring to the differences in coefficients. We have simplified the sentence as suggested by the referee.

Line 284: "Given the same carbon and oxygen contents, the partitioning behaviors of  $C_9H_{12}O_x$  and  $C_9H_{14}O_x$  should be similar to each other."

Line 445: Do the authors have any levoglucosan calibrations from the same time period as these experiments to help connect these bulk sensitivities to those already published? If so, I strongly encourage the authors to include that value

We have added information regarding the EESI-TOF sensitivity towards levoglucosan to Sections 2.4.

Line 203: "The  $RF_x^*$  towards dry, polydisperse levoglucosan (i.e. 1,6-anhydro- $\beta$ -D-glucose, Sigma-Aldrich, > 99%) particles nebulized from aqueous solutions ranged from 7730 ± 2130 ions s-1 ppb-1, or roughly 1170 ± 320 ions s-1 of  $[C_6H_{10}O_5+Na]^+$  observed per  $\mu g m^{-3}$  of levoglucosan particles sampled."

**References**

Bi, C., Krechmer, J. E., Frazier, G. O., Xu, W., Lambe, A. T., Claflin, M. S., Lerner, B. M., Jayne, J. T., Worsnop, D. R., Canagaratna, M. R. and Isaacman-VanWertz, G.: Coupling a gas chromatograph simultaneously to a flame ionization detector and chemical ionization mass spectrometer for isomer-resolved measurements of particle-phase organic compounds, Atmos. Meas. Tech., 14(5), 3895–3907, doi:10.5194/amt-14-3895-2021, 2021.

Claflin, M. S., Pagonis, D., Finewax, Z., Handschy, A. V., Day, D. A., Brown, W. L., Jayne, J. T., Worsnop, D. R., Jimenez, J. L., Ziemann, P. J., de Gouw, J. and Lerner, B. M.: An in situ gas chromatograph with automatic detector switching between PTR- and EI-TOF-MS: isomerresolved measurements of indoor air, Atmos. Meas. Tech., 14(1), 133–152, doi:10.5194/amt-14-133-2021, 2021.

Duncianu, M., David, M., Kartigueyane, S., Cirtog, M., Doussin, J. F. and Picquet-Varrault, B.: Measurement of alkyl and multifunctional organic nitrates by proton-transfer-reaction mass spectrometry, Atmos. Meas. Tech., 10(4), 1445–1463, doi:10.5194/amt-10-1445-2017, 2017.

Heinritzi, M., Simon, M., Steiner, G., Wagner, A. C., Kürten, A., Hansel, A. and Curtius, J.: Characterization of the mass-dependent transmission efficiency of a CIMS, Atmos. Meas. Tech., 9(4), 1449–1460, doi:g, 2016.

Hodzic, A., Aumont, B., Knote, C., Lee-Taylor, J., Madronich, S. and Tyndall, G.: Volatility dependence of Henry's law constants of condensable organics: Application to estimate depositional loss of secondary organic aerosols, Geophys. Res. Lett., 41(13), 4795–4804, doi:10.1002/2014GL060649, 2014.

Leglise, J., Müller, M., Piel, F., Otto, T. and Wisthaler, A.: Bulk Organic Aerosol Analysis by Proton-Transfer-Reaction Mass Spectrometry: An Improved Methodology for the Determination of Total Organic Mass, O:C and H:C Elemental Ratios, and the Average Molecular Formula, Anal. Chem., 91(20), 12619–12624, doi:10.1021/acs.analchem.9b02949, 2019.

Pagonis, D., Price, D. J., Algrim, L. B., Day, D. A., Handschy, A. V., Stark, H., Miller, S. L., de Gouw, J., Jimenez, J. L. and Ziemann, P. J.: Time-Resolved Measurements of Indoor Chemical Emissions, Deposition, and Reactions in a University Art Museum, Environ. Sci. Technol., 53(9), 4794–4802, doi:10.1021/acs.est.9b00276, 2019.

Qi, L., Chen, M., Stefenelli, G., Pospisilova, V., Tong, Y., Bertrand, A., Hueglin, C., Ge, X., Baltensperger, U., Prévôt, A. S. H. and Slowik, J. G.: Organic aerosol source apportionment in Zurich using an extractive electrospray ionization time-of-flight mass spectrometer (EESI-TOF-MS) - Part 2: Biomass burning influences in winter, Atmos. Chem. Phys., 19(12), 8037–8062, doi:10.5194/acp-19-8037-2019, 2019.

Sekimoto, K., Li, S. M., Yuan, B., Koss, A., Coggon, M., Warneke, C. and de Gouw, J.: Calculation of the sensitivity of proton-transfer-reaction mass spectrometry (PTR-MS) for organic trace gases using molecular properties, Int. J. Mass Spectrom., 421, 71–94, doi:10.1016/j.ijms.2017.04.006, 2017.

Stefenelli, G., Pospisilova, V., Lopez-Hilfiker, F. D., Daellenbach, K. R., Hüglin, C., Tong, Y., Baltensperger, U., Prévôt, A. S. H. and Slowik, J. G.: Organic aerosol source apportionment in Zurich using an extractive electrospray ionization time-of-flight mass spectrometer (EESI-

TOF-MS) – Part 1: Biogenic influences and day–night chemistry in summer, Atmos. Chem. Phys., 19(23), 14825–14848, doi:10.5194/acp-19-14825-2019, 2019.

Yuan, B., Koss, A. R., Warneke, C., Coggon, M., Sekimoto, K. and De Gouw, J. A.: Proton-Transfer-Reaction Mass Spectrometry: Applications in Atmospheric Sciences, Chem. Rev., 117(21), 13187–13229, doi:10.1021/acs.chemrev.7b00325, 2017.